# A biological camera that captures and stores images directly into DNA

Cheng Kai Lim[1,2,3,4,5], Jing Wui Yeoh[1,4], Aurelius Andrew Kunartama[1,4], Wen Shan Yew [1,2,3] & Chueh Loo Poh [1,4] ✉

The increasing integration between biological and digital interfaces has led to heightened interest in utilizing biological materials to store digital data, with the most promising one involving the storage of data within defined sequences of DNA that are created by de novo DNA synthesis. However, there is a lack of methods that can obviate the need for de novo DNA synthesis, which tends to be costly and inefficient. Here, in this work, we detail a method of capturing 2-dimensional light patterns into DNA, by utilizing optogenetic circuits to record light exposure into DNA, encoding spatial locations with barcoding, and retrieving stored images via high-throughput next-generation sequencing. We demonstrate the encoding of multiple images into DNA, totaling 1152 bits, selective image retrieval, as well as robustness to drying, heat and UV. We also demonstrate successful multiplexing using multiple wavelengths of light, capturing 2 different images simultaneously using red and blue light. This work thus establishes a 'living digital camera', paving the way towards integrating biological systems with digital devices.

DNA is a key biomaterial that forms the basis of biological life on earth. It serves as the storage of genetic information that encodes for the multitude of proteins which fulfil various functions of life. Due to this ability to act as a storage of information, along with its simple, repeating code of 4 nucleotides, ATCG, DNA has been proposed and subsequently explored as a form of digital information storage[1,2]. This is due to its extreme density (petabytes per gram[3]), longevity (DNA has been retrieved from samples millions of years old[4]), and continued relevance to study due to its underpinning of biological life. These properties have led to a recent boom in developing different workflows[5–9] for converting digital data into DNA and vice versa, which has been increasingly relevant and urgent due to the impending shortage of silica necessary for manufacturing storage devices required to accommodate our projected data storage requirements[3].

Current DNA storage workflows largely rely on in vitro synthesis of DNA strands, which are costly and require complex instrumentation[10–12]. Errors in the synthesis process are also common. While there have been substantial advances in accelerating this process by developing enzymatic synthesis methods[13,14], miniaturizing electrochemical synthesis[15–17], or developing more robust encoding methods[18–20], DNA synthesis remains a bottleneck in the adoption of DNA as a data storage medium. There is thus significant interest in developing ways of encoding information into DNA that can either supersede or circumvent DNA synthesis in its current form.

The abundance of DNA in living cells have been considered as a potential source of DNA that can be encoded with information via the use of molecular biology tools and biological systems. Utilizing living cells as a way to incorporate and record external signals such as the presence of chemicals or electrical inputs into DNA have also been demonstrated, with recording systems ranging from recombinases[21–23] to CRISPR-based modifications[24–26]. Most notably, recent work has showcased the storage of 72 bits of information directly into the

[1]Synthetic Biology for Clinical and Technological Innovation, National University of Singapore, 28 Medical Drive, Singapore 117456, Singapore. [2]Synthetic Biology Translational Research Programme, Yong Loo Lin School of Medicine, National University of Singapore, 14 Medical Drive, Singapore 117599, Singapore. [3]Department of Biochemistry, Yong Loo Lin School of Medicine, National University of Singapore, 8 Medical Drive, Singapore 117597, Singapore. [4]Department of Biomedical Engineering, College of Design and Engineering, National University of Singapore, Singapore, Singapore. [5]Integrative Sciences and Engineering Programme (ISEP), NUS Graduate School, National University of Singapore, Singapore, Singapore. ✉e-mail: poh.chuehloo@nus.edu.sg

genomic DNA of living cells via external electrical input signals[9], as well as the use of DNA structural barcodes that bind to the M13mp18 bacteriophage DNA scaffold to encode information[7,27]. However, such systems have not been able to accurately capture spatial information, due to the lack of mechanisms that can simultaneously encode spatial information as well as external input signals. Furthermore, existing encoding systems also tend to utilize low-throughput chemical processes, which severely limits scalability.

In this paper, we present a workflow that allows the direct capture of both spatial information and input signals via light into DNA itself as a means to store digital information such as images into DNA. We utilize a blue light-responsive recombinase system that responds to the presence or absence of blue light as an external signal, and subsequently records that response into DNA itself via site-specific DNA editing. To enable the encoding of spatial information along with the recorded signal, we implemented a barcoding scheme to enable differentiation of individual wells containing cells with records of light exposure, thereby 'digitizing' the recorded image and allowing for deconvolution upon retrieval of the DNA sequences by sequencing. As a result, this recombinase-based DNA recorder, coupled with the aforementioned barcoding scheme that can preserve spatial information, enables the direct capture of both spatial information as well as input signals via light into DNA itself. While previous work has demonstrated the usage of light-responsive systems that can capture light and display this input as a corresponding light output, akin to an analogue camera[28], our process instead creates a biological analogue to a digital camera which we term as 'BacCam'. As the DNA encoded with different images can be pooled and stored together, we further characterize this workflow by demonstrating random access and multiplexing, and utilize outlier detection and reassignment techniques, along with unsupervised clustering algorithms from the field of machine learning to accurately reconstruct the stored data. Taking advantage of the multiplexing capabilities offered by light, we further layer an orthogonal red light system with the blue light system to allow encoding of 2 images simultaneously, increasing the scalability and density of the workflow and enabling multicolor image capture. Utilizing molecular biology methods, optogenetics, barcoding techniques, and biological systems, we thus provide a framework for the integration of biological and digital interfaces.

## Results

### BacCam enables the direct-to-DNA storage and retrieval of images

Light has the distinct advantages of being cheap, massively parallelizable, rapid, highly programmable, and easily multiplexed, with little effort or cost needed to scale or generate patterns of increasing complexity. There is therefore an interest in utilizing light as a patterning mechanism in biology, such as in photolithographic DNA synthesis[17], as well as optogenetic circuits for multiple purposes such as bioproduction or controlling cellular behavior on a spatiotemporal level[29–31]. We therefore sought a way to utilize light as an input to encode information into DNA. We hypothesized that cells containing optogenetic circuits that can record the presence or absence of light within DNA can be perceived as analogous to a digital camera that captures images via light exposure and records said exposure in a digital format. We thus utilized an Opto-Cre-Vvd recombinase system, whereby a Cre-Lox recombinase protein was engineered to be light inducible by splitting the recombinase and attaching photodimers that bring the split protein together upon exposure to blue light[21]. The recombinase, upon activation, excises a predefined section of DNA that are flanked by LoxP sites, resulting in an alteration in the sequence. We posited that this alteration is analogous to the encoding of a bit. To determine whether a bit has been encoded, upon sequencing, the total number of reads that possess the excised LoxP sequence is compared to the total number of reads with the full LoxP sequence. A high ratio of excised reads to unexcised ones would correspond to a bit state of '1' being encoded, while the opposite would correspond to a bit state of '0'.

We then posited that a 96-well plate, with each well being appropriately barcoded, and containing cells with the Opto-Cre-Vvd and LoxP genetic circuits, would be analogous to a digital camera with image sensors each containing individual pixels that have their own unique identifier, storing information corresponding to light exposure. To scale the one-bit digital drive within each cell to that of a multipixel image, a population of *Escherichia coli* (*E. coli*) bacteria containing the above-mentioned circuit was spatially separated in individual wells of a black, clear-bottomed 96-well plate, upon which a predefined pattern of 465 nm wavelength light was projected from below.

The projected pattern is then preserved by the addition of unique oligonucleotide barcodes, which we term as 'well-codes', that bind at the region preceding the LoxP recording site. The nucleotide sequence of the well-code is mapped to predefined spatial locations, with said mappings saved in a separate table, thereby resulting in the linkage of the spatial location of the recording bacteria in each isolated well along with the recorded digital drive that each bacteria possesses. The addition of the well-code to the recording sequence is conducted via PCR. Further oligonucleotide sequences were then appended via PCR to prepare the individual samples for Next-Generation Sequencing (NGS). Subsequently, all samples are then pooled together for storage, which obviates the need for preserving the original well configuration as the unique well-codes have tied this configuration with the recorded bit state. This results in a pool of DNA oligonucleotide sequences that stores the information of the image that was captured (Fig. 1). In practice, this barcoding system enables the capture of 2-dimensional images, where each pixel's location is represented by a well-code, and the bit state of the pixel is determined by the presence or absence of the DNA excision site. This resulting data can be stored robustly in dried form on a benchtop at room temperature, or frozen in a −20 °C fridge.

To retrieve the image, sequencing of the pooled DNA was then conducted. The resulting sequencing data generated was then deconvoluted by determining the total number of sequences retrieved for each barcode, and the bit state determined by the ratio of sequences lacking the region between the LoxP sites to sequences containing the region. These ratios indicate the proportion of DNA that were excised compared to ones that were intact. High ratios would thus indicate that a larger proportion of DNA was excised, which would then imply that light exposure was 'recorded' and so recording a '1' signal, and vice versa for lower ratios, which would then record a '0' signal. The threshold determining the ratio that demarcates assignment of a '1' or a '0' signal was derived using clustering methods that compare ratios across each plate. This deconvolution was done for all well-codes used, and the resulting bit-well-code pairs were then reassembled back, according to a previously stored mapping table of well-codes to spatial locations, to form a digitized version of the projected image.

Utilizing the above-mentioned process, we have successfully demonstrated the viability of various aspects of the workflow detailed above. We demonstrated the recording of a predefined pattern into DNA and retrieved it via sequencing (Fig. 1). Deconvolution of the sequenced reads was done by mapping the well-code back to its spatial location, counting the number of excised and intact LoxP reads corresponding to each well-code, and determining the bit state of each well via the above-mentioned ratio of excised to intact LoxP reads.

### Multiple images can be stored in a complex pool and retrieved with high accuracy

We subsequently tested the multiplexing capability of our workflow, generating multiple images and pooling them together, and deconvoluting each image from the DNA sequencing data

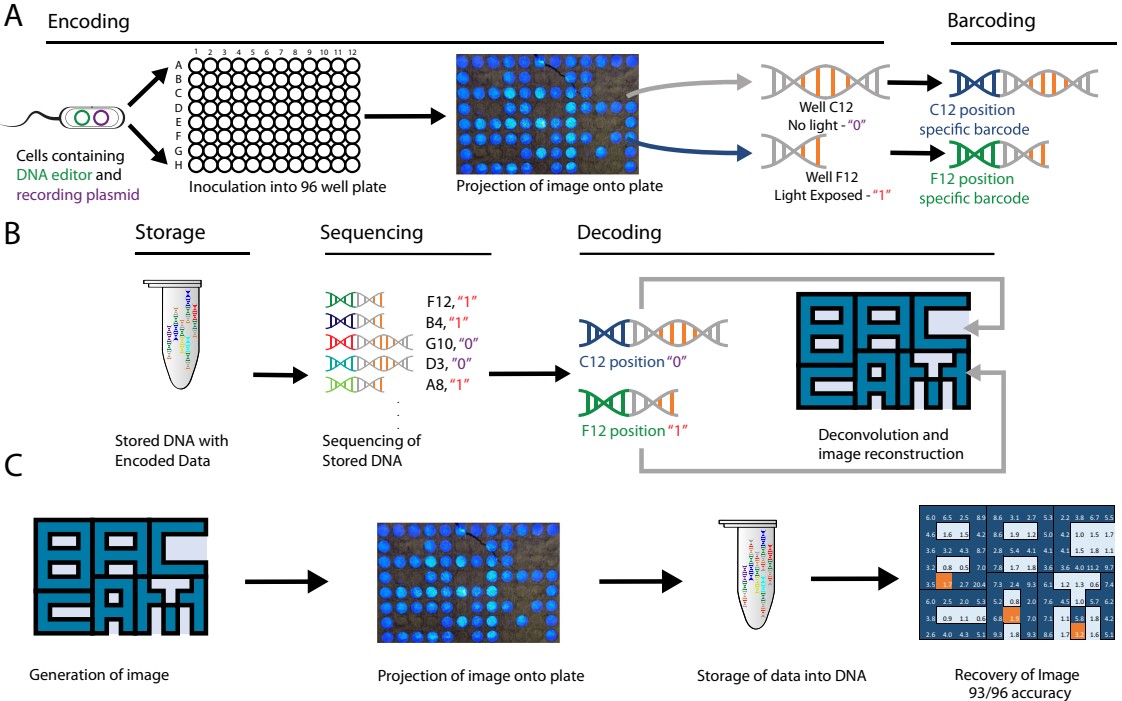

**Fig. 1 | Workflow of BacCam. A** Encoding of light into bacteria and barcoding for preservation of spatial information. **B** Sequencing and decoding of stored DNA back into original image. **C** Initial proof of concept of BacCam using the 'BACCAM' pattern. Successful reconstruction of the image was done with 93/96 accuracy. Source data are provided as a Source Data file.

generated via sequencing of said pool. We hypothesized that each full image can also be barcoded with its own 'meta-barcode' that separates images from one another in a heterogenous DNA pool of multiple images, despite using the same well-codes that separate individual wells, allowing for multiplexing. We thus incorporated this second layer of indexing using indexing barcodes provided by Illumina. Each 96 well plate, after the addition of the initial well-codes, has the same 2 indexing barcodes appended on the 5′ and 3′ end of the sequence in each well. These barcodes are combinatorial in nature, thus increasing the possible number of total images that can be stored together.

To determine if this second indexing layer can be used to deconvolute multiple images from the same sample, we exposed 5 unique patterns to 5 different 96-well plates, thereby creating a set of 5 images (Fig. 2A). 4 of these patterns (NUS, SYNCTI, BacCam, and Smiley) illustrate the ability of the system to capture images. The last pattern, 'Heloo wo{|d!' serves to illustrate the ability of BacCam to encode information (such as letters and symbols represented in 8-bit ASCII code format) by allocating each well as a bit and encoding the information in a 96-well format. Hence, each column (8 wells) will encode one ASCII code. In this case, BacCam can also serve to encode any sort of information, as long as it is projected with the appropriate pattern.

We then added the same set of well-codes to each of them via PCR, before extending each image with indexing barcodes. Each image used a different combination of indexing barcodes, which were then saved in a mapping table that links the unique index sequences with its corresponding image. These indexed images were then pooled together into the same pool of DNA. We then determined if a mixed, heterogenous pool of DNA can be sequenced as a whole library, with the individual images deconvoluted from the total raw reads that are produced by sequencing. We demonstrate that a mixed pool of at least 5 different images can be deconvoluted and reconstructed with an accuracy of at least 90% for all images, showcasing the multiplexing capability of our workflow (Fig. 2B).

We subsequently tested the ability of our workflow to implement random access of images, whereby each image can be selectively amplified and decoded without the need to sequence the entire mixed DNA pool. We hypothesized that designing primers that selectively bind to the indexed sequences corresponding to the desired image would be sufficient for random access, adequately amplifying each individual 'pixel' associated with the image, while avoiding amplification of sequences that do not possess the selected index. To test the hypothesis, we demonstrated that all images (with accuracy > 80%) in a mixed pool of 5 can be selectively amplified and accessed by using the corresponding indexing primers and conducting a touchdown PCR to ensure precise binding and amplification of desired sequences[32] (Fig. 2C), thus obviating the need to sequence the entire pool of DNA. Taken together, these results show that multiple images can be easily tagged, stored together and subsequently demultiplexed by using a simple indexing process that complements existing NGS workflows.

## Information can be accurately reconstructed from small initial quantity of samples

To determine the minimum amount of DNA required to reconstruct images accurately, we conducted a series of dilution experiments. We diluted a mixed pool of samples progressively, with each dilution being tenfold less than the previous. Sequencing of each dilution for a particular image for deconvolution and reconstruction demonstrated that images were accurately reconstructed at dilutions of a hundred-fold from the initial concentration (2.66 nM). At dilutions of a 1000-fold (2.66 pM), individual reads for each barcode numbered in the low single digits, with many having no reads at all. As a result, the fidelity of the image drops significantly (Fig. 3A). For large scale multiplexing of images, the concentration of DNA representing a single image drops as the number of images in each pool increases. This dilution assay thus provides us with an approximate proxy for the number of images that can be concurrently retrieved in a sequencing run, assuming a fixed volume of the pooled library was sequenced, and sequencing coverage remained the same. The results imply that the number of different

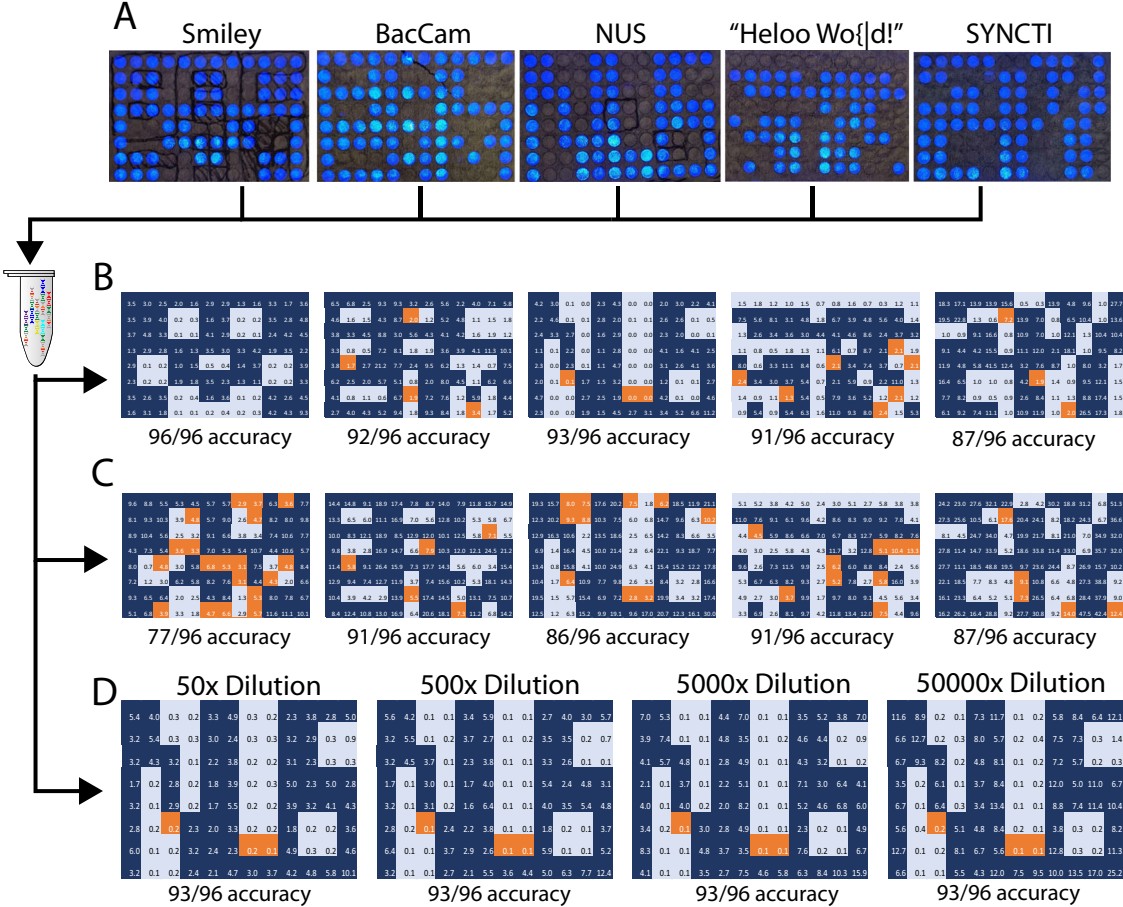

**Fig. 2 | Storage and retrieval of multiple images in a single complex pool.** Red boxes indicate wells with allocated bits that do not correspond with the projected image. **A** Projected patterns that are stored with the BacCam process. Images are projected on different plates, barcoded individually with index sequences corresponding to each image, and resulting products are pooled together into a single tube. **B** Retrieved images from a single sequencing run of a mixed pool, deconvoluted with the corresponding indexes of each image. **C** Retrieved images from multiple sequencing runs, whereby random access of each image was conducted from the mixed pool by using specific primers to amplify desired images before sequencing. **D** Reconstruction of selectively amplified single images from mixed libraries of increasing dilutions, demonstrating robustness of the amplification technique in selecting specific images from an ever-decreasing initial amount of DNA. Source data are provided as a Source Data file.

images that can be stored in a pool and retrieved in a single run is between 100 and 1000. Increasing the number of distinct images that can be retrieved this way would therefore require an increase in sequencing coverage and/or increasing concentrations of each image.

However, we also hypothesized that PCR amplification of diluted reads via random access might be sufficient to overcome the above-mentioned limit without an increase in sequencing coverage, saving costs while still allowing for multiplexed storage in the same pool. We therefore conducted serial dilutions on the mixed pool of images and utilized indexing primers for random access of selected images via PCR, amplifying the diluted DNA and sending it for sequencing. We demonstrated that the NUS image was selectively amplified with the N701 and N501 indexes (index sequences in Supplementary Table 1), from a 50× dilution of an initial concentration of 17.98 nM to 0.36 nM, up to a dilution of 50,000× (0.36 pM), showcasing the robustness of image retrieval from a complex pool of 5 images, and demonstrating consistent re-amplification of images from a small quantity of initial samples (Fig. 2D). Attempts to retrieve the other images contained in the mixed library after selective amplification were unsuccessful, demonstrating the capabilities of targeted amplification of selective images. It is also apparent that at lower dilutions, there is an increasing skew in terms of the ratio values−suggesting that amplification from minute amounts of sample tends to dramatically increase the contrast of reads that are excised compared to unexcised ones.

## Images stored are durable in varying environmental conditions

We further explored the limits of BacCam by subjecting it to a battery of physical challenges. One way to improve the density and stability of DNA is to dry it, due to the hydrolytic activity of water molecules on the phosphate backbone. This also has the effect of reducing volume, resulting in a higher overall density. To test the viability of our workflow for drying, we spun dried a volume of DNA corresponding to the 'Smiley' image before rehydration and sequencing. We showed that images in dried form could be successfully retrieved with zero loss in accuracy compared to liquid, frozen images (Fig. 3B).

We also subjected a DNA sample with the 'Heloo wo{|d!' image to accelerated aging experiments. This was done by comparing identical samples, both in aqueous form with one at room temperature (RT) and another that was kept in a 60 °C oven, both for a duration of 1 week. We also exposed a freshly thawed frozen sample to UV light for 1 hr. It was previously demonstrated that data encoded in dried DNA stored at a temperature of 70 °C for a duration of 7 days was not able to be retrieved, and that duration was equivalent to storing DNA at 9.4 °C for 2000 years[33]. We demonstrated a successful retrieval of said image from all samples, with no loss of information despite varying

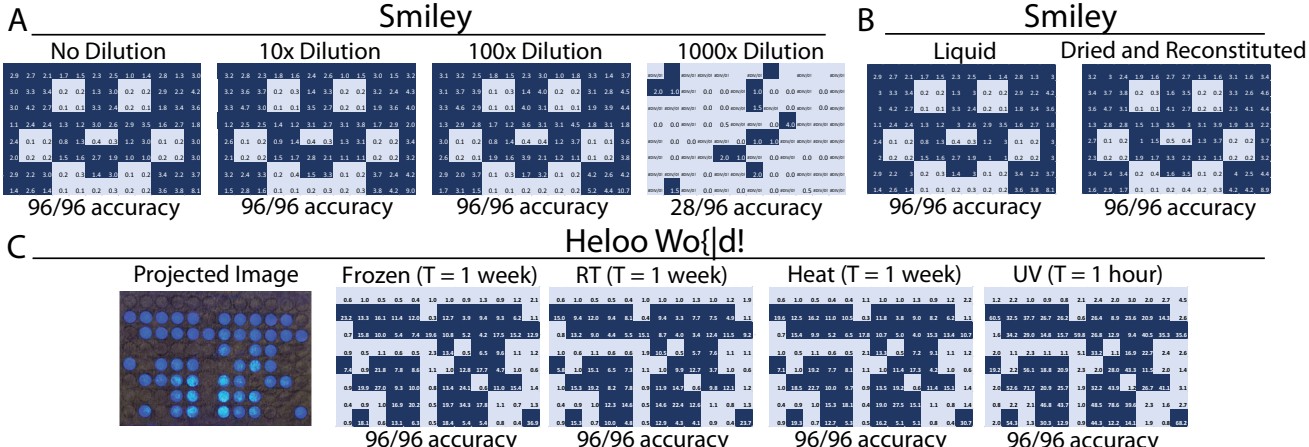

**Fig. 3 | Testing the boundaries of BacCam. A** Dilution experiments demonstrating the theoretical maximum information capacity of BacCam. Images can be successfully retrieved up to a dilution of 100× from the original fully amplified pool, with 1000x dilution having significantly higher number of errors due to the low number of reads. **B** No loss in information for 'Smiley' stored in both liquid and dried form. **C** No loss of information in 'Heloo wo{|d!' stored in varying conditions for extended periods of time. Source data are provided as a Source Data file.

conditions, showcasing the inherent robustness and redundancy of the workflow (Fig. 3C).

## Optimization of image deconvolution with computational methods

Upon receipt of sequencing data, proportions of excised and unexcised DNA was calculated and allocated to appropriate wells. To determine the correct threshold that will allow for accurate separation of recorded light signals, we utilized a manual thresholding process based on foreknowledge of the encoded image. This method was used to assess the accuracy of the overall workflow in storing a projected image into DNA. While this method suffices due to foreknowledge of the pattern encoded, to enable a true information encoding and retrieval system, a method that can reconstruct a potential image solely from the sequenced data would be necessary. We theorized that computational clustering methods such as unsupervised machine learning techniques could be utilized to automatically identify the distinct groups/clusters. To preprocess the sequencing data before implementing the clustering methods, we applied an outlier detection technique also known as unsupervised anomaly detection to detect the outliers located at the low-density regions[34]. We proceeded to reassign the outliers with the nearest values detected from the inliers. This outlier detection and reassignment data preprocessing demonstrated improvement in removing small low-density clusters and enhancing the edges among larger clusters. Post-processing techniques were also implemented to satisfy the edge cases of having fully 'ON' or 'OFF' images and to perform cluster grouping to retrieve these final binary state images. The entire automated workflow for image deconvolution is outlined in Fig. 4A.

We tested multiple clustering methods under different parameter settings on the existing datasets, and compared the accuracy of the automated clustering methods with manual thresholding based on foreknowledge of the existing image (Table 1). The results showed that the OPTICS clustering method and 3-component Gaussian Mixture Model (GMM) produced clusters that most closely emulated manual thresholds, with near perfect accuracy (Supplementary S2 and S3). We tested the results acquired using GMM at different phases of the workflow (Fig. 4B), showing how each phase contributes to building an accurate end result, and demonstrated the robustness of the automated workflow by showcasing all the successfully deconvoluted images with high accuracy (>0.9) (Fig. 4C, Supplementary S2).

## Multiplexing with multiple colors of light

One significant advantage of using light is the ability to multiplex in a simple fashion with the addition of different light wavelengths. We redesigned a red light sensitive Cre recombinase (pBbS5a-RLCre) initially developed in yeast systems to add the red wavelength to the existing blue light BacCam workflow[35]. To allow for differentiation between the two light systems in the workflow, we designed another recording plasmid (pBbAW4k-Spacer1Barcoding-loxP-TT-loxP-ho1-pcyA) that contained another intermediate barcode, thus differentiating edits caused by red light exposure to those caused by blue light exposure (Fig. 5A, Supplementary S1). We also designed a programmable light illumination device (OptoBox, Supplementary S8) to project multiple colors of light in the same well. We show that a co-culture of the blue light and red light-responsive bacteria in the same well is able to sense multiple wavelengths of light at the same time, and encode 2 separate images simultaneously. Images were encoded in two separate ways. One way involved the overnight projection of a blue light pattern along with a different red light pattern in an alternating fashion, with each pattern being projected for 10 min before switching. Another way involved projecting both images simultaneously (Fig. 5B). We demonstrate the successful encoding and retrieval of two different images with different wavelengths of light, with both methods of projection being viable approaches, with a minimum accuracy of more than 90% for each image encoded (Fig. 5C, Supplementary S5−S7). These results demonstrate high orthogonality of both light systems (Supplementary S9).

## Discussion

In this work, we have proposed and demonstrated an alternative workflow for DNA data storage—a direct capture of images into DNA that is analogous to the creation of a digital camera. We demonstrated scaling of this workflow using a single color light (blue) wavelength to five 96 bit images, totaling 60 bytes, and showed that the theoretical limits of this system potentially scales to more than one hundred 96 bit images in a single heterogenous pool. We have also shown the ability for random access of individual images, even from a highly dilute concentration of 50,000x lesser than the initial concentration, and the reliable recovery of data from diluted, heat treated, UV-exposed and dried pools. We have implemented computational pipelines that serve to deconvolute the encoded information and correct for errors in a reliable fashion. To scale this workflow beyond a single wavelength of light, we incorporated an additional wavelength of light, doubling the

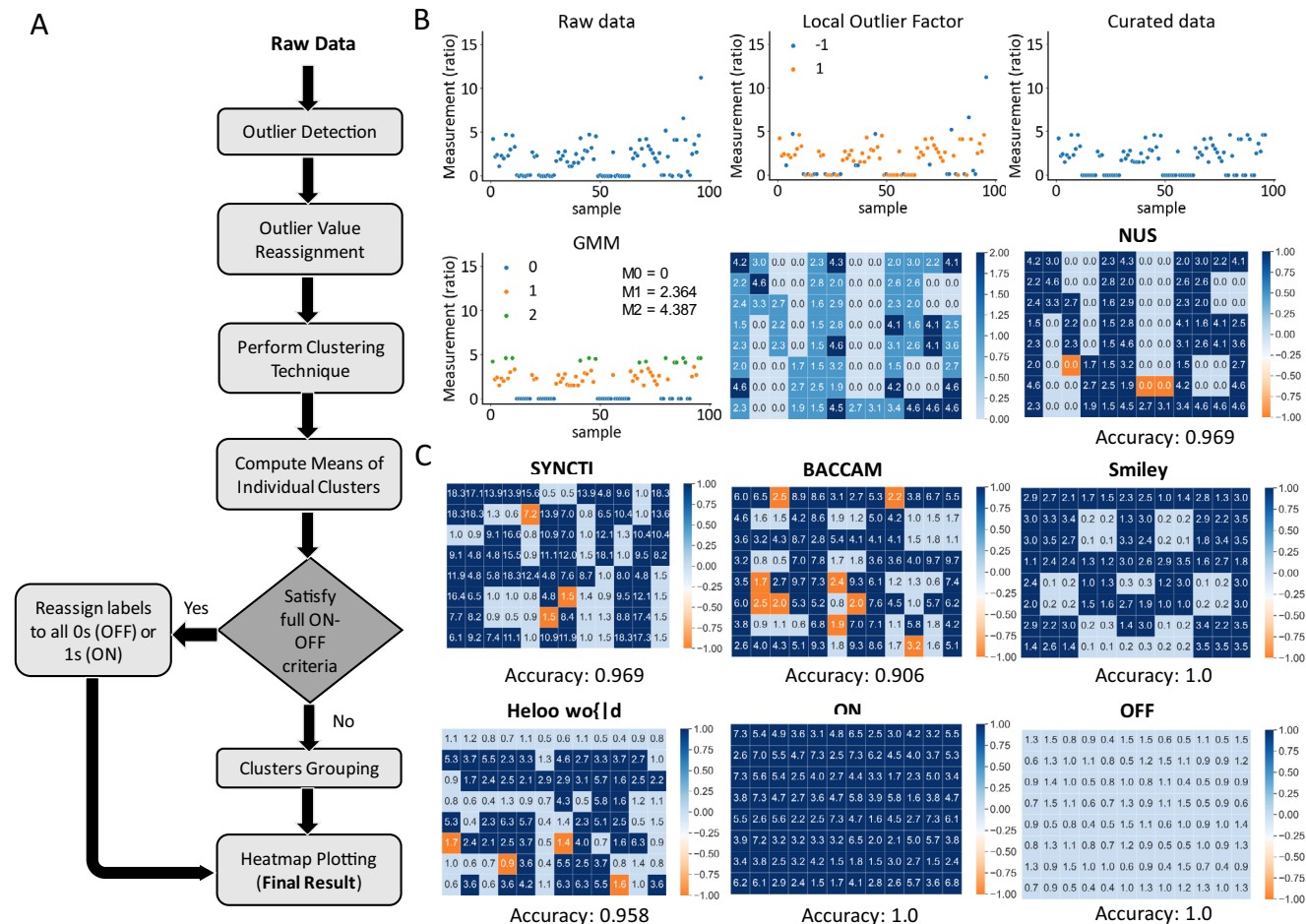

**Fig. 4 | Developing a machine learning clustering-based workflow for automated image deconvolution. A** An automated workflow incorporating outlier detection and reassignment method, machine learning unsupervised clustering technique, full ON-OFF criteria assessment, and cluster grouping for automated image deconvolution. **B** An example implemented using GMM to demonstrate the results acquired at different phases. The dots in the graphs represent the raw data or the curated data from the 96 wells. The '1' and '−1' obtained from the Local Outlier Factor denote the inliers and outliers detected respectively from the raw data. The curated data were acquired after reassigning the outliers to the nearest values of inliers. The M0–M2 indicate the computed mean values of the individual clusters for full ON−OFF criteria assessment. The deconvoluted image was then plotted based on the three clusters and after clusters grouping into the final binary state image ('0': light blue; '1': dark blue) with 3 error bits (orange). **C** Validation of the automated workflow on other patterns including full 'ON' and 'OFF' datasets.

## Table 1 | Comparison of methods used for automated thresholding

| Methods | K-Means | | OPTICS | DBSCAN | | Gaussian Mixture Model (GMM) | | Manual thresholding |
|---|---|---|---|---|---|---|---|---|
| Settings/Patterns | 2 clusters | 3 clusters | | eps = 0.4 | eps = 0.2 | 2 component | 3 component | |
| NUS | 0.885 | 0.969 | 0.969 | 0.969 | 0.969 | 0.969 | 0.969 | 0.969 |
| SYNCTI | 0.646 | 0.896 | 0.969 | 0.969 | 0.844 | 0.969 | 0.969 | 0.958 |
| BACCAM | 0.708 | 0.865 | 0.906 | 0.875 | 0.823 | 0.708 | 0.906 | 0.969 |
| SMILEY | 0.792 | 1.0 | 1.0 | 1.0 | 1.0 | 0.771 | 1.0 | 1.0 |
| Heloo wo{|d! | 0.760 | 0.938 | 0.938 | 0.938 | 0.948 | 0.958 | 0.958 | 0.958 |
| OFF | 1.0 | 1.0 | 1.0 | 1.0 | 1.0 | 1.0 | 1.0 | – |
| ON | 1.0 | 1.0 | 1.0 | 1.0 | 1.0 | 1.0 | 1.0 | – |

amount of data that can be stored in a single, simultaneous capture and demonstrating the multiplexing potential of the system. This system further expands the boundaries of the field outside of pre-existing DNA synthesis and sequencing workflows.

We have utilized living cells as the encoder for DNA data storage in this work. This offers several advantages over de novo in vitro DNA synthesis-based data storage formats. Living cells are an ever-renewing source of DNA, making it cost effective to produce at scale. Cells also possess a multitude of systems that can

interface with DNA, such as transcription factor binding proteins, and are also able to respond to a multitude of stimuli ranging from chemical to light and electrical means that can also be recorded into DNA in vivo. Furthermore, in BacCam, the writing step is done in parallel, as each bit is encoded at the same time. As such, latency of writing is greatly reduced.

While living cells have been previously used as encoders for storing images, BacCam differs primarily from existing work by combining spatial addressability along with optogenetic systems

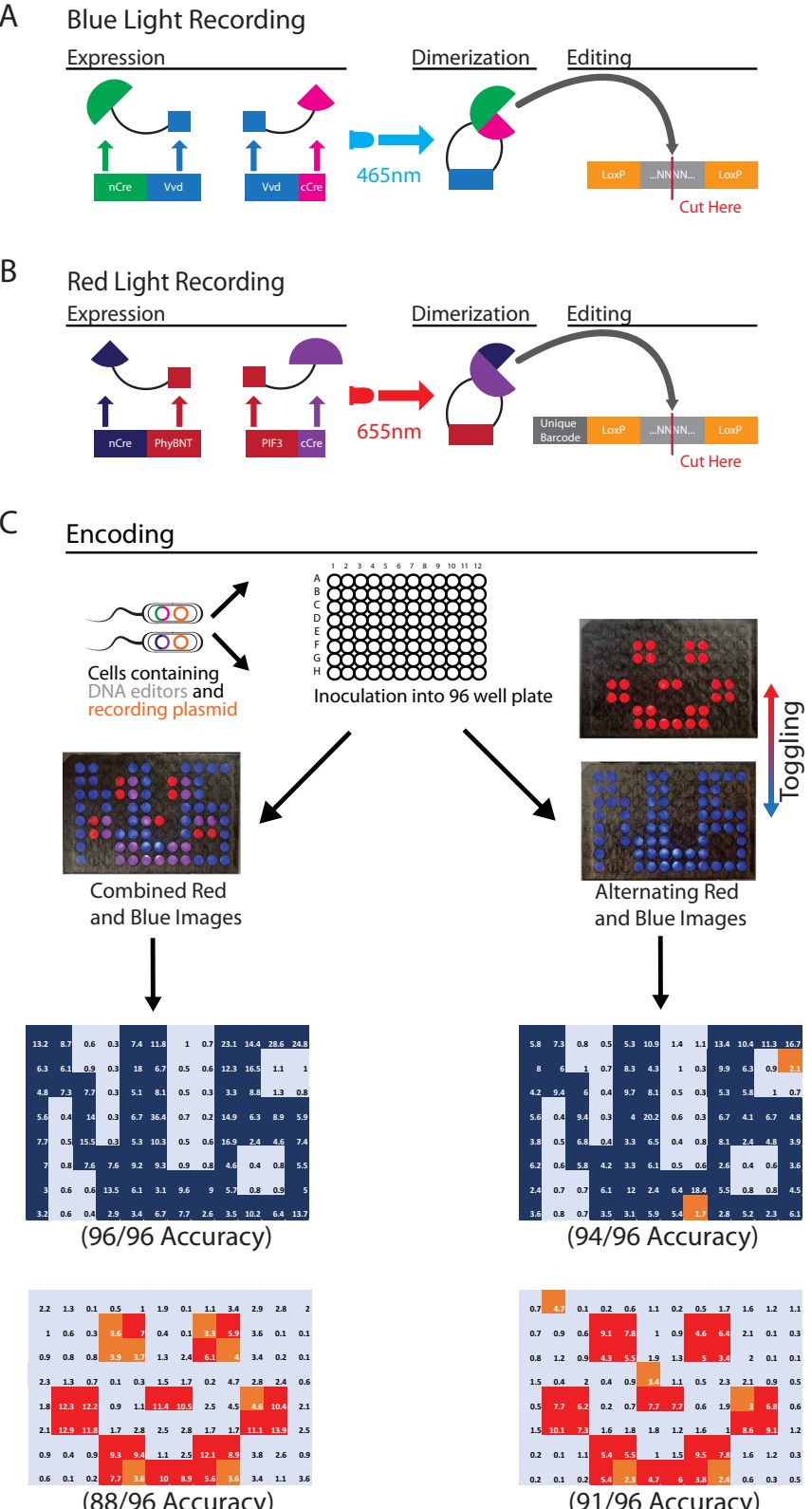

**Fig. 5 | Multiplexing the BacCam workflow using multiple colors of light.**
**A** Design of a co-culture of engineered *E. coli* with blue and red light sensitive recombinases respectively. The blue light Opto-Cre-Vvd system and the redesigned red light recombinase recording system were separately transformed into *E. coli*. **B** Experimental workflow of the co-culture. Both resulting strains of *E. coli* were co-cultured together on the same plate, and exposed to either alternating blue and red light projections, or simultaneous projections of both blue and red light. **C** Images obtained after sequencing and deconvolution, separating red and blue with the red light specific barcode and subsequently subjecting it to our deconvolution algorithm workflow based on 3-component Gaussian Mixture Model (GMM). Source data are provided as a Source Data file.

to digitally encode information directly into DNA. This is in contrast to previous work such as that of Shipman et al.[24,25], in which image information was encoded via de novo DNA synthesis before insertion into the bacterial genome. Furthermore, the multiplexing ability of light allows us to utilize different wavelengths of light to encode additional layers of information, providing significant flexibility in encoding and offering potential avenues for capacity expansion.

The resolution of images encoded is a function of the number of unique well-codes generated, as well as the physical separation and isolation of individually addressable bacterial populations. As a proof of concept, we have thus far utilized 96-well plates for convenience and cost. However, we envision that subsequent implementations of the workflow can be further miniaturized and scaled up to 384, 1536 or even higher resolution systems, leveraging advances in liquid handling devices, microchips[36,37], as well as microfluidic setups[38]. The robustness of PCR amplification in retrieving information from a complex pool of oligonucleotides has been demonstrated earlier, and indicates that there is still significant potential capacity for data storage. Implementation of various PCR or DNA assembly methods that do not require thermocycling for appending well-codes and indexing barcodes can also increase the throughput of the process while reducing the footprint and complexity of the workflow[39,40]. Advances in barcode multiplexing methods, as well as metadata tagging, can potentially lead to a searchable image database, as proposed by previous groups[40,41].

While we utilize light-responsive recombinases in our workflow as the main interface between light and DNA recording, it is by no means the only DNA writing system that can be used. Other advances in DNA writing and editing tools can also serve to act as the intermediary for the conversion of light to information stored in DNA, and possess further advantages such as increased orthogonality. This could lead to an increase in density and the expansion in the types of images that can be stored.

As the field of DNA data storage continues to progress, there is an increasing interest in bridging the interface between biological and digital systems. Our work showcases further applications of DNA data storage that recreate existing information capture devices in a biological form, providing the basis for continued innovation in information recording and storage.

## Methods

### Capturing of blue light patterns into DNA
Two plasmids, pBbAW4k-loxP-TT-loxP-mRFP1 (pLoxP) and pBbE5a-Opto-Cre-Vvd-2 (pOptoCre) were used. pBbAW4k-loxP-TT-loxP-mRFP1 was a gift from Mary Dunlop (Addgene plasmid # 134405; http://n2t.net/addgene:134405; RRID:Addgene_134405) and pBbS5a-Opto-Cre-Vvd-2 was a gift from Mary Dunlop (Addgene plasmid # 160400; http://n2t.net/addgene:160400; RRID:Addgene_160400). These plasmids were transformed together into XL1-Blue *E. coli* chemically competent cells. pLoxP contains two LoxP sites within the plasmid in the same orientation, with an intervening sequence containing 2 terminator sequences. pOptoCre contains a genetic circuit that utilizes the Lac promoter to control production of the Opto-Cre-Vvd-2 construct. The transformed strain was inoculated into a culture tube with 5 ml of LB medium supplemented with 100 mg/ml of ampicillin and 50 mg/ml of kanamycin and grown overnight in a shaking incubator at 37 °C, aerobically. The culture was diluted 1:100 into a fresh culture tube with 10 ml of LB medium supplemented with 100 mg/ml of ampicillin and 50 mg/ml of kanamycin and induced with 0.1 μM of IPTG for 2 h in a 37 °C shaking incubator. Eighty microliters of refreshed and induced cells were aliquoted into each well of a 96-well, black and flat clear-bottomed plate. A light pattern was created by selectively blocking out the bottoms of wells with aluminum foil, and the plate was exposed to light from the bottom with a blue LED light pane (HQRP). The plate was

incubated atop the device overnight at room temperature to ensure sufficient exposure to light. However, this duration can be shortened to 30 min if necessary as demonstrated in Supplementary S4.

### Barcoding of DNA and library prep for illumina sequencing
Ninety-six different PCR reactions were set up. Each reaction well contains 0.5 μl of a specific known well-code at a concentration of 10 μM that is mapped to the $X−Y$ coordinates of that well. The unique barcoding sequences were generated computationally using the 'DNABarcodes' R package in Bioconductor (version 1.28.0)[42]. One microliter of cells from each well of the 96-well plate was then used for the PCR reactions. PCRs were performed using Taq DNA polymerase (NEB) with each reaction containing 2 μl of 10X ThermoPol buffer (NEB), 0.1 μl of 100 μM of each forward and reverse primer pair (IDT), 0.2 μl of Taq DNA polymerase (NEB), 0.2 μl of 20 mM dNTP mix (Bio-Basic), 1 μl of cells, and topped up with ddH$_2$O for a total reaction volume of 20 μl. Individual reactions were run on 1.2% agarose gels for product verification. Appending of adapter ends for sequencing was also done using PCR, with 0.2 μl of each barcoded mix from the previous reaction added into a reaction containing 0.2 μl Taq DNA polymerase (NEB), 2 μl of 10X ThermoPol buffer (NEB), 0.1 μl each of 100 μM inner sequencing index primers (IDT), 0.3 μl of 100 μM outer sequencing index primers (IDT), 0.2 μl of 20 mM dNTP mix (BioBasic), and topped up with ddH$_2$O to a total volume of 20 μl for each reaction. Products were subsequently verified on agarose gel before aliquoting 10 μl from each reaction and pooled together into a single tube. Two hundred microliters of this mixed pool was sent for NovaSeq dual index paired-end sequencing with an external sequencing service provider (NovogeneAIT).

### Deconvolution of raw reads into images
The resulting raw data generated was deconvoluted with appropriate index sequences and separated accordingly by the external sequencing provider. Subsequent analysis of the data was done using an in-house R script (code provided in the Supplementary Software file) (R version 4.2.2[43], Bioconductor version 3.16[44], Biostrings version 2.66.0[45], ShortRead version 1.56.1[46] and stringr version 1.5.0[47] packages). Briefly, the script derived the total number of reads, the number of reads with excised intervening LoxP sequences, and number of reads with full LoxP sequences, that were linked to each unique well-code. To determine whether that unique barcoded population was exposed to light or not, the ratio of truncated to full reads was determined. This ratio is calculated for each well in the entire plate, and serves to define the bit state of the well. A higher ratio indicates a larger proportion of truncated reads, which implies that a large number of the population of cells that were barcoded was exposed to light and vice versa. To reconstruct the images, each the aforementioned ratio was converted to either 1 or 0, with the threshold between both determined with GMM clustering to define the two clusters and the threshold set via the boundary between the two clusters with the lowest variance score. This is then linked back to the well-code, which was then mapped to its predefined spatial location, thus creating an image with a pixel depth of 1 bit and a resolution of 96 pixels. Accuracy of the final image was defined as the number of wells that were appropriately demarcated as 1 or 0 as compared to the original projected pattern. Differentiating between two colors of light was done by calculating ratios of sequences specific to the blue light recording plasmid as well as the red light recording plasmid separately from one another.

### Random access of images
Primers specifically designed for random access with index sequences targeted at selected indexes were synthesized and used to enable random access. Eight PCR reactions with the same conditions were set up to generate sufficient DNA for sequencing. 1 μl of template from the

mixed pool was used for each reaction totaling 50 μl, and a touchdown PCR with decreasing annealing temperatures from 65 to 55 °C for eight cycles and constant annealing temperatures for 20 cycles was conducted, using specific random access primers that bind to desired image to be amplified. Resulting samples were appended with Illumina adapter sequences and sent for sequencing. Diluting experiments were conducted by taking 1 μl of template from the mixed pool and diluting in 9, 99, and 999 μl of PBS before taking 1 μl from each dilution for PCR as per the above protocol. N701 and N501 indexing primers selective for the 'NUS' pattern were used for this particular random access dilution.

### Environmental challenge experiments

A pool of DNA containing the 'Smiley' pattern was used for drying and resuspension experiments. Twenty microliters of the pool was spun dry using a centrifuge, and subsequently resuspended in 50 μl ddH$_2$O before being submitted for sequencing.

For environmental challenge experiments, 200 μl of the 'Heloo wo{|d!' pool was kept in varying conditions in a cryotube, wrapped in aluminum foil. Conditions included frozen in −20 °C, at room temperature, and in a 60 °C oven, all for 7 days. One last sample was kept in a 1.5 ml Eppendorf tube which was then exposed to UV light for 1 hour. Samples were then sent for sequencing.

### Dilution experiments

Two forms of dilution experiments were conducted. One involved diluting a pool of DNA containing the 'Smiley' pattern 3 consecutive times, at a 10 times dilution each, with each dilution being sent for sequencing. The other involved diluting a mixed pool of DNA with multiple samples 3 consecutive times, with each dilution being a 10 times dilution as described above. Random access was then conducted on each dilution as detailed above before sending the amplified products for sequencing.

### Clustering of deconvoluted images

The clustering workflow was developed and implemented in Python 3[48] (version 3.11.1) (code provided in the Supplementary Software file). Four unsupervised clustering techniques (K-means algorithm[49], DBSCAN[50], OPTICS[51], and Gaussian Mixture Model[52]) were adopted from scikit-learn (version 1.2.0) machine learning packages under cluster and mixture modules[53]. The unsupervised outlier detection algorithm was implemented using Local Outlier Factor (LOF) from scikit-learn neighbors module. The parameter 'n_neighbors' (blue light = 20; red light = 10) was tuned accordingly to improve the deconvolution accuracy through better reassignment of outliers near to the cutoff threshold for better clustering performance (Supplementary S6 and S7). The cutoff threshold for the full 'ON' and 'OFF' was determined from the means computed from clusters from all existing datasets. The dataset would be classified as full 'ON' or 'OFF' when the lowest mean of clusters is above the cutoff threshold, or the highest mean is below the cutoff threshold. To generate the final binary '0' or '1' images, cluster grouping was executed for clustering techniques that identified more than 2 clusters to group them into two clusters designated as 'ON' and 'OFF' states. For the blue light patterns, the second and higher mean clusters were grouped into a single 'ON' cluster whereas the initial cluster with the lowest mean would be considered as 'OFF' states. Meanwhile, for the red light patterns, the two clusters with the lower mean were considered as a single 'OFF' cluster instead whereas the rest of the clusters would be considered as 'ON' states due to differences in the distribution of the raw data.

### Development of the red light bacterial recombinase system

Two plasmids were designed for the red light bacterial recombinase system. pBbS5a-RLCre, a red light sensitive split Cre-recombinase based off the L-SCRaMbLE system previously developed in yeast[35] was designed, codon-optimized for *E. coli* and synthesized (Twist Bioscience). pBbAW4k-Spacer1Barcoding-loxP-TT-loxP-ho1-pcyA, a recording plasmid, was also designed, consisting of LoxP recognition sites, a unique preceding barcode (Spacer1Barcoding), as well a ho1-pcyA component that produces phycocyanobilin necessary for the function of the red light split recombinase. Both plasmids were transformed simultaneously into XL1 Blue *E. coli* chemically competent cells.

### Design and construction of a custom light illumination device (OptoBox)

To test the co-culture of red and blue light-responsive *E. coli* under various blue and red light patterns, a 96-well programmable LED device was developed (design illustrated in Supplementary S8). This device is powered and controlled by an Arduino Uno Microcontroller and includes 8 strips of 12 Adafruit Neopixel SMD 5050 RGB LEDs totalling 96 individual LEDs. Each of these LEDs corresponds to each well of the 96-well plate. The overall construction of the device comprises a 96-tip pipette tip container. These pipette tip containers comprise two separable layers stacked on top of one another. This setup allows the LED strips to be immobilized between the layers. The wells of the pipette tip container also prevent residual light spillover from the LEDs, resulting in spatial isolation when coupled with a clear-bottomed black 96-well plate.

For our application, these hollow tip-holes are used to fit and secure the SMD 5050 LEDs in place by fitting the LED chips into the underside of each of these corresponding holes. This allows each individual LED to be held in place through a friction fit, with the diodes that make up the SMD 5050 LED centered to the wells of the 96-well plate. The LED strips were soldered to wires supplying power and electrical control. Each strip's data-wire was connected to an individual Arduino Uno data pin, allowing individual control of each LED. Programming of the LEDs was done using the available FastLED Arduino library.

### Capture of multicolor images with a co-culture

Strains containing the red light and blue light recombinase systems were inoculated separately into culture tubes with 5 ml of LB medium supplemented with 100 mg/ml of ampicillin and 50 mg/ml of kanamycin and grown overnight in a shaking incubator at 37 °C, aerobically. These cultures were diluted 1:100 into fresh culture tubes with 10 ml of LB medium supplemented with 100 mg/ml of ampicillin and 50 mg/ml of kanamycin and induced with 0.1 μM of IPTG for 2 h in a 37 °C shaking incubator. 40 μl of each culture were aliquoted into each well of a 96-well, black and flat clear-bottomed plate. A red (620 nm) and a blue (465 nm) light pattern was created by the in-house custom-built light illumination device (OptoBox), and either projected periodically for 10 min in an alternating fashion overnight, or projected simultaneously overnight, both from the bottom of the plate and at equal intensities. Barcoding, sequencing and deconvolution follows the same procedure as detailed in the preceding sections.

### Statistics and reproducibility

No statistical method was used to predetermine sample size. No data were excluded from the analyses. The experiments were not randomized. The Investigators were not blinded to allocation during experiments and outcome assessment.

### Reporting summary

Further information on research design is available in the Nature Portfolio Reporting Summary linked to this article.

## Data availability

The sequencing data generated in this study have been deposited in a Figshare repository at: https://doi.org/10.6084/m9.figshare.22678534.

The same data have also been deposited in the NCBI Sequence Read Archive under the PRJNA970212 BioProject at https://www.ncbi.nlm.nih.gov/bioproject/PRJNA970212, with the individual accession numbers available in the supplementary file under Supplementary Table 2. Source data are provided with this paper. Barcodes and primers used in this work are detailed in the Source Data file.

## Code availability

The code used for deconvolution of reads as well as clustering is available in the Supplementary Software file.

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

## Acknowledgements

We thank the Yew and Poh Lab members for constructive criticism critical to the research. This work was supported by Ministry of Education, Singapore, under its AcRF Tier 2 Grant (MOE-T2EP30221-0014). C.K.L. is funded by the NUS Integrative Sciences and Engineering Programme.

## Author contributions

C.K.L. and C.L.P. developed the initial concept. C.K.L. performed the experiments and analyzed the results under the supervision of C.L.P. and W.S.Y. C.K.L., J.W.Y. and C.L.P. developed and constructed the computational pipeline for image reconstruction. A.A.K. designed and developed the OptoBox device. C.K.L., J.W.Y., A.A.K. and C.L.P. wrote the manuscript with input from all authors.

## Competing interests

The authors declare no competing interests.
