## [Peer Review File · Nature Communications]

Reviewers' Comments:

Reviewer #1:

Remarks to the Author:

Review of "A biological camera that captures and stores images directly into DNA"

This manuscript presents a method to directly convert 2D arrays of binary data (principally demonstrated using images) into DNA using a light-controlled recombinase-based system. Light is used to activate the recombinase, which results in excision of a synthetic DNA segment in-vivo. Sequencing is then used to readout the presence/absence of the DNA segment within a population of cells – which can be translated to binary 0/1 bits. Individual bits can be written in parallel by spatially isolating individual populations of bacteria in 96-well plates. Bits are addressed using DNA indexes, allowing the different bits to be pooled together following the writing process.

From one perspective, this is certainly an interesting approach to using light to control enzymatic DNA synthesis for digital data storage, which is likely a top contender for what will ultimately be the best synthesis route some time in the future. As the authors say, light is a very powerful method of controlling molecular mechanisms; it's cheap, controllable in both time and space, etc. Pair it with enzymatic synthesis and then you have a potentially winning combination.

On the other hand, there are a couple of points in the broader approach this paper takes that mitigate my excitement for its technological and conceptual contribution. First, this paper claims to be the "first living camera" – I think it deserves to be pointed out a number of earlier works that have used biological cells as "cameras", for example see early and recent works by the Voigt lab, Tabor lab, and others, that used biofilms as biological cameras wherein cells were programmed to sense and respond to patterns of light (eg images), albeit, the current manuscript may be the first to write the information directly into DNA, as opposed to storing the information at the level of transcriptional regulation. Second, the majority of the paper is essentially of re-demonstration of earlier papers in the field of DNA data storage, namely: pooling and random access of multiple files using PCR and DNA-based addresses; stress testing of DNA-based data stability (ie robustness to temperature, dehydration, etc), dilution/recovery experiment, direct writing of bits-to-DNA in-vivo (eg Wang lab, Church lab, work using electricity and or chemical induction), etc. This re-hashing does not contribute substantially to the impact of the work unfortunately. Perhaps the authors could consider how to increase the density and scalability of their writing system, maybe by multiplexing with different wavelength of light?

Some smaller critiques:

- For the dilution experiments, it would be great if the authors could use units of mass and/or concentration, rather than "Nx Dilution" (which is arbitrary).
- The authors claim that image resolution is limited by the number of unique well-codes. I think this is not quite right; their current system is likely limited by the physical constraints of their experimental setup, specifically having to spatially isolate and individually-address with light, unique populations of bacteria (1 population per bit of data). This scheme, and its limitations in general, make it hard to see this approach become scalable.
- The colors in dot plots (eg Fig 4), it's not clear to me what those represent.

Reviewer #2:

Remarks to the Author:

This paper describes a light controllable DNA writer for DNA storage using blue light-inducible recombinase that can excise the DNA in each well of a 96 well plate based on the light pattern. Modified DNA was then extracted and mixed in a pool, and then the information can be requested through DNA sequencing on-demand. The robustness and application of this DNA writer were characterized and verified under different storage conditions. Based on the considerations mentioned later in the comment, I believe that the paper can be published with further revision.

Major concern:

I appreciate the inventive idea of using light-inducible recombinase for DNA storage. While other work used light-inducible recombinase for diagnostics, therapeutics, and cell sensing, this paper combines synthetic genetic circuits with digital data storage. This method does increase the efficiency and decrease the cost of writing memory though avoiding the process of DNA synthesis. I also appreciate their effort in mixing the data of 5 different 96 well plates and extracting each image individually from the mixed pool. Although it comes with the cost of losing some of the data in each image, it does propose a strategy to minimize the storage space by combining different DNA information into one single pool and then correctly extracting them individually when requested.

Many limitations of this method are still needed to be addressed and improved.

1. Experiments should be performed to characterize how the exposure time and blue light strength can affect the recombinase genetic circuit and information storage. The current method is performed by inducing the bacteria under blue light overnight, which is a very time-consuming data writing process. Researchers should determine what is the shortest exposure time that is required to write data and make the whole process more efficient.
2. In Figure 3C, there is data lost, 4/96 of information, after 7 days at room temperature. Experiments should be performed to optimize the storing conditions and find out what causes the loss of 4 bits of DNA information, which should be quite stable under room temperature.
3. There are other techniques that can record images into DNA. It isn't exactly clear how this approach is better than others, e.g., the CRISPR system developed by Shipman et al (church lab). Therefore, it would be helpful to demonstrate the unique advantages of collecting spatial information, and to prove that the blue light-inducible recombinase allows for broader applications that other recording systems cannot achieve.

Minor errors:

There are some typos and grammar issues in the paper. For example, the aluminum was misspelled into aluminium, and dashes were missing between two words in several locations such as ninety-six, well-being, etc. Examples of grammar issues include mistakenly using "a small amount of initial samples" rather than "a small number of initial samples".

We are thankful for the substantive and insightful comments that have been provided by the reviewers to improve the quality of our work. We have addressed these comments individually and have revised our manuscript accordingly and believe that our work has been greatly improved as a result. Please find our responses to the comments below.

Reviewer #1 (Remarks to the Author):

Review of “A biological camera that captures and stores images directly into DNA”

This manuscript presents a method to directly convert 2D arrays of binary data (principally demonstrated using images) into DNA using a light-controlled recombinase-based system. Light is used to activate the recombinase, which results in excision of a synthetic DNA segment in vivo.

Sequencing is then used to readout the presence/absence of the DNA segment within a population of cells – which can be translated to binary 0/1 bits. Individual bits can be written in parallel by spatially isolating individual populations of bacteria in 96-well plates. Bits are addressed using DNA indexes, allowing the different bits to be pooled together following the writing process.

From one perspective, this is certainly an interesting approach to using light to control enzymatic DNA synthesis for digital data storage, which is likely a top contender for what will ultimately be the best synthesis route some time in the future. As the authors say, light is a very powerful method of controlling molecular mechanisms; it's cheap, controllable in both time and space, etc. Pair it with enzymatic synthesis and then you have a potentially winning combination.

We thank the reviewer for their positive comments.

On the other hand, there are a couple of points in the broader approach this paper takes that mitigate my excitement for its technological and conceptual contribution. First, this paper claims to be the “first living camera” – I think it deserves to be pointed out a number of earlier works that have used biological cells as “cameras”, for example see early and recent works by the Voigt lab, Tabor lab, and others, that used biofilms as biological cameras wherein cells were programmed to sense and respond to patterns of light (eg images), albeit, the current manuscript may be the first to write the information directly into DNA, as opposed to storing the information at the level of transcriptional regulation. Second, the majority of the paper is essentially of re-demonstration of earlier papers in the field of DNA data storage, namely: pooling and random access of multiple files using PCR and DNA-based addresses; stress testing of DNA-based data stability (ie robustness to temperature, dehydration, etc), dilution/recovery experiment, direct writing of bits-to-DNA in-vivo (eg Wang lab, Church lab, work using electricity and or chemical induction), etc. This re-hashing does not contribute substantially to the impact of the work unfortunately. Perhaps the authors could consider how to

increase the density and scalability of their writing system, maybe by multiplexing with different wavelength of light?

We thank the reviewer for their insightful comments. As was astutely pointed out by the reviewer, it is indeed true that earlier works have used biofilms as biological cameras. What we would like to emphasize is that the difference in this work lies in it being the first demonstration of directly capturing images into DNA, without the need of an intermediate conversion step. To continue the analogy of cameras, previous work has created analogue cameras capturing images in the form of film, while our work is proposing the first digital camera, capturing images directly into 'digital DNA'. We believe that this is a conceptual leap that is not immediately apparent, and that this is an initial proof of concept that serves to demonstrate the viability of this workflow. We have amended our introduction to bring this concept across in a clearer fashion and thus distinguish our work from earlier works that used biofilms as biological cameras.

We appreciate that the corpus of work in the field has demonstrated various properties of DNA data storage in the form of pooling/random access. We believe that our workflow, given its inherently different encoding properties, necessitates the re-demonstration of what was previously done to show that our workflow is comparable or better in terms of robustness, accuracy, and extensibility to existing workflows. We believe that this re-demonstration showcases the strength of our workflow in terms of lowered cost, higher throughput, and high-density, effective data retrieval.

We acknowledge that further extension of this work could entail multiplexing or density improvements, which opens up potential areas for future research and development.

Some smaller critiques:

-For the dilution experiments, it would be great if the authors could use units of mass and/or concentration, rather than "Nx Dilution" (which is arbitrary).

Thank you for the comment - we have made the changes as described. The initial starting concentration of the undiluted DNA was 17.98nM as quantified by qPCR.

-The authors claim that image resolution is limited by the number of unique well-codes. I think this is not quite right; their current system is likely limited by the physical constraints of their experimental setup, specifically having to spatially isolate and individually-address with light, unique populations of bacteria (1 population per bit of data). This scheme, and its limitations in general, make it hard to see this approach become scalable.

The reviewer is indeed correct that the physical constraints of the experimental setup is a limitation to the workflow. We have earlier discussed this limitation in the discussion section. Fundamentally, as pointed out by the reviewer, these physical constraints are analogous to the ever-present question of miniaturization and scalability that occurs in all plate-based assays. The advancements that have been made in miniaturizing, multiplexing and scaling such assays, as detailed in work such as μ SCALE¹, FemDA², or high-throughput sequencing chips, have successfully overcome physical limitations in 96 well formats, which thus provides engineering means to overcome these limitations. We have amended the manuscript to emphasize these points.

-The colors in dot plots (eg Fig 4), it's not clear to me what those represent.

Apologies for the confusion. We have amended the description for Fig. 4 and supplementary figures to provide more details on the plots and the colors within them.

Reviewer #2 (Remarks to the Author):

This paper describes a light controllable DNA writer for DNA storage using blue light-inducible recombinase that can excise the DNA in each well of a 96 well plate based on the light pattern. Modified DNA was then extracted and mixed in a pool, and then the information can be requested through DNA sequencing on-demand. The robustness and application of this DNA writer were characterized and verified under different storage conditions. Based on the considerations mentioned later in the comment, I believe that the paper can be published with further revision.

Major concern:

I appreciate the inventive idea of using light-inducible recombinase for DNA storage. While other work used light-inducible recombinase for diagnostics, therapeutics, and cell sensing, this paper combines synthetic genetic circuits with digital data storage. This method does increase the efficiency and decrease the cost of writing memory though avoiding the process of DNA synthesis. I also appreciate their effort in mixing the data of 5 different 96 well plates and extracting each image individually from the mixed pool. Although it comes with the cost of losing some of the data in each image, it does propose a strategy to minimize the storage space by combining different DNA information into one single pool and then correctly extracting them individually when requested.

We thank the reviewer for their kind comments.

Many limitations of this method are still needed to be addressed and improved.

1. Experiments should be performed to characterize how the exposure time and blue light strength can affect the recombinase genetic circuit and information storage. The current method is performed by inducing the bacteria under blue light overnight, which is a very time-consuming data writing process. Researchers should determine what is the shortest exposure time that is required to write data and make the whole process more efficient.

The exposure time required to write the data depends on the activity of the light-sensitive recombinase, which has been well characterized by the original developers of the Opto-Cre system. To further address this, we have conducted a set of experiments showcasing the encoding of the image over time for half-hour intervals with the data as shown in the new supplementary Figure S4. From the results, we thus demonstrated that at 30 minutes, images can be encoded, albeit with some errors, and 2 hours are sufficient for robust image encoding, with 96/96 pixels accurately reconstructed. For other experiments, we set the exposure overnight to maximize the extent of editing. Further engineering of the recombinase to make it more responsive to light could be conducted in future iterations.

2. In Figure 3C, there is data lost, 4/96 of information, after 7 days at room temperature. Experiments should be performed to optimize the storing conditions and find out what causes the loss of 4 bits of DNA information, which should be quite stable under room temperature.

Thank you for the astute observation. We have repeated the experiment, adjusting the set up to generate more robust results. The revised experiments now include an initial sequencing of a freshly encoded sample to determine the original image (along with any initial encoding errors) stored, and subsequently, the generated DNA was exposed to 4 different conditions (room temperature, heated at 60°C, frozen at -20 °C - all for 1 week, along with exposure to UV light for 1 hour for the frozen sample), to show the robustness of the DNA encoded. The results show that although slight errors occurred in the initial encoding step, the same image was preserved with no change over all conditions tested, demonstrating that the encoded DNA is robust. As can be seen, the workflow is not perfect and does not always produce error-free images. Nonetheless, this work provides the foundation for further optimization of the workflow to reduce the error rates and improve robust retrieval.

3. There are other techniques that can record images into DNA. It isn't exactly clear how this approach is better than others, e.g., the CRISPR system developed by Shipman et al (church lab). Therefore, it would be helpful to demonstrate the unique advantages of collecting spatial information, and to prove that the blue light-inducible recombinase allows for broader applications that other recording systems cannot achieve.

Thank you for the comment. Currently, other systems such as that of Shipman et. al. encoded images by the conversion of the image into *de novo* synthesized DNA sequences. These synthesized oligonucleotides are then incorporated into the genome of cells, thereby 'recording'

images into DNA. However, these images must have already existed prior to the encoding step, and this process is therefore more accurately described as transferring information from a digital file into DNA to be stored in bacteria. This process, which necessitates the usage of *de novo* synthesized oligonucleotides, also result in significant costs.

However, in our work, we have developed a way to capture said information in the form of light straight into DNA, without the intermediate step of encoding and synthesizing the DNA, and without the need for a pre-existing digital file encoded in DNA. This is analogous to the usage of a digital camera that captures light and saves that as an image in a digital form. As such, we believe that this demonstration thus shows the power of spatial addressing to enable this sort of direct capture, which, when coupled with optogenetic recombinase systems, enables an easy, cost-efficient and effective method of encoding information directly into DNA without the need for *de novo* DNA synthesis. This method is also broadly applicable with other spatial addressing and light-sensitive recording systems, serving as a robust basis for further development. We have further clarified the novelty of our approach from existing ones in the discussion section.

Minor errors:

There are some typos and grammar issues in the paper. For example, the aluminum was misspelled into aluminium, and dashes were missing between two words in several locations such as ninety-six, well-being, etc. Examples of grammar issues include mistakenly using “a small amount of initial samples” rather than “a small number of initial samples”.

Thank you for the corrections. We have made the appropriate language changes as suggested.

1. Chen, B. *et al.* High-throughput analysis and protein engineering using microcapillary arrays. *Nat. Chem. Biol.* **12**, 76–81 (2016).
2. Zhang, Y. *et al.* Accurate high-throughput screening based on digital protein synthesis in a massively parallel femtoliter droplet array. *Sci. Adv.* **5**, eaav8185 (2019).

Reviewers' Comments:

Reviewer #1:

Remarks to the Author:

The authors have adequately addressed my concerns.

Reviewer #2:

Remarks to the Author:

The authors have adequately addressed my comments. I have no further comments.